

# SIX4 promotes metastasis via activation of the PI3K-AKT pathway in colorectal cancer

Guodong Li, Fuqing Hu, Xuelai Luo, Junbo Hu and Yongdong Feng

Cancer Research Institute, Tongji Hospital, Huazhong University of Science and Technology, Wuhan, Hubei, China

## ABSTRACT

**Background:** Several studies report aberrant expression of sine oculis homeobox (SIX) homolog family members during cancer development and progression. SIX4 participates in organ development, such as myogenesis and neurogenesis. However, the expression and clinical implication of SIX4 in colorectal cancer (CRC) remains unclear.

**Methods:** The SIX4 expression levels in colorectal patients were assessed in nine different human cancer arrays and compared using patient survival data. SIX4 expression was silenced in two cell culture lines for invasion and wound healing assessment. Finally, bioinformatics assessments ascertained the pathways impacted by SIX4.

**Results:** SIX4 was upregulated in The Cancer Genome Atlas CRC cohort and other gene expression omnibus (GEO) cohorts. In addition, SIX4 expression significantly correlated with lymph node metastasis and advanced Tumor Node Metastasis (TNM) stages. Moreover, SIX4 overexpression was related to unfavorable prognosis in CRC patients. Silencing SIX4 inhibited CRC cell metastasis by surpressing AKT phosphorylation.

**Discussion:** SIX4 is upregulated in CRC and can be used as a prognosis biomarker.

Corresponding author
Yongdong Feng,
ydfeng@tjh.tjmu.edu.cn

## INTRODUCTION

Colorectal cancer (CRC) is one of the most prevalent malignant neoplasms with both incidence and mortality ranked third in the world (*Jemal et al., 2011*). Despite gradual improvement in its prognosis through innovative therapeutic strategies, many CRC patients still die. The CRC morbidity has steadily increased as lifestyles have changed in China. Thus, it is imperative to identify new CRC biomarkers to improve the predictive value for CRC prognosis, which could enhance our understanding of carcinogenesis and tumor progression.

The sine oculis homeobox (SIX) homolog family is comprised of six members, SIX1, SIX2, SIX3, SIX4, SIX5, and SIX6 (*Hu, Mamedova & Hegde, 2008*; *Seo et al., 1999*). There are two conserved domains shared among the SIX family members. The SIX domain participates in protein interactions, whereas the HD is involved in DNA binding (*Elhashash et al., 2011*). SIX4, also known as AREC3, contains 760 amino acids and

localizes to the nucleus. SIX4 participates in organ development, such as myogenesis and neurogenesis. As a member of SIX family, SIX4 contains one DNA-binding motif, suggesting that the protein has DNA-binding capacity. SIX4 can function as a regulatory factor by binding the Na+/K+-ATPase subunit gene (*Kawakami et al., 1996*).

Several studies report abnormal SIX family member expression during cancer formation and progression. SIX1 participates in invasion and proliferation during tumor development in non-small cell lung cancer (*Xia et al., 2014*). *Ono et al. (2012)* demonstrate that SIX1 inhibits the expression of E-cadherin and induces epithelial-to-mesenchymal transition by increasing ZEB1 expression in CRC. It promotes cell proliferation by reactivating the cell cycle-related protein cyclin D (*Kong et al., 2014*; *Li et al., 2013*). More importantly, it is closely linked to a poor clinical prognosis in cancer patients (*Kong et al., 2014*). Furthermore, SIX2 promotes breast cancer metastasis through downregulating E-cadherin (*Wang et al., 2014*). However, SIX3 has an opposite effect on the clinical outcome of lung adenocarcinoma patients. Upregulation of SIX3 in lung cancer cells results in inhibition of cell proliferation and migration (*Mo et al., 2013*). SIX4 expression closely correlates to the progress and prognosis of esophageal squamous cell carcinoma (*Wei et al., 2013*).

Despite recent efforts to understand the role of SIX4 as a transcriptional factor, little is known about the roles and expression of SIX4 in CRC. Therefore, the mechanism of SIX4 in CRC deserves further studies. In this study, we clarify SIX4 expression in CRC and then analyze the relationship between SIX4 expression and clinicopathological features. Moreover, the function and mechanism of SIX4 in CRC cell metastasis was revealed by bioinformatics analysis and in vitro assays.

## MATERIALS AND METHODS

### Bioinformatics analysis

Level 3 HiSeq RNASeq data were downloaded from The Cancer Genome Atlas (TCGA) website (*Zhu et al., 2009*) for 380 cancer samples. In addition, eight different human expression arrays (GSE5206 (*Kaiser et al., 2007*), GSE9348 (*Yi et al., 2010*), GSE20916 (*Skrzypczak et al., 2010*), GSE17536 (*Smith et al., 2010*), GSE6988 (*Ki et al., 2007*), GSE14333 (*Jorissen et al., 2009*), GSE20842 (*Gaedcke et al., 2010*), and GSE39582 (*Marisa et al., 2013*)) and the corresponding clinical information was downloaded from the gene expression omnibus (GEO) website. The selection of GEO datasets was carried out as follows: (1) According to the CRC datasets list of Oncomine database (*Rhodes et al., 2004*), we obtained the datasets containing cancer and normal (or adjacent) tissues in the GEO database; (2) The selection of CRC datasets containing survival information in GEO database was carried out according to following keywords "colorectal cancer" or "colon cancer" and "prognosis" or "prognostic" or "outcome" or "survival" or "recurrence" or "relapse" and "Homo sapiens". After setting Entry type and time as "Series" and "31/12/2015," there were 171 datasets passing filtering. In addition, we set the conditions as follows: number (>100), Expression profiling type (Expression profiling by array and no SuperSeries), clinical information (OS), Manufacturer (Affymetrix).

Differences between SIX4 expression in different tissues were calculated by the Student's $t$-test. In order to determine the prognosis significance of SIX4 in CRC patients, survival differences were validated through the Kaplan–Meier method. In order to select the relationships between SIX4 and AKT-related genes, we applied Pearson's correlation analysis to the mRNA expression data. R package survival was used to analyze the relationship between SIX4 expression and survival from TCGA and other CRC cohorts. The fold change between different tumor progression stages and SIX4 expression levels were calculated. Gene ontology (GO) provides a framework for biology computation models. GO provides ontologies to describe gene functions and relationships. It classifies genes according to molecular function, cellular components, and biological processes. The list of all genes correlated with SIX4 was submitted to the database for annotation, visualization, and integrated discovery (DAVID) online tools for GO and Kyoto encyclopedia of genes and genomes (KEGG) pathway enrichment analysis (*Huang, Sherman & Lempicki, 2008*).

Gene set enrichment analysis (GSEA) software was acquired from the Broad Institute at MIT. TCGA dataset, including 380 CRC samples, was analyzed by GSEA. The "hallmark gene sets" were used for running GSEA. The phenotype labels were generated according to the expression level of SIX4. $P < 0.05$ and FDR $Q < 0.25$ were set as the default parameters to generate enrichment results.

## Clinical tissue samples

Fresh CRC tissues and paired adjacent nontumor colorectal tissues were obtained from CRC patients at Tongji Hospital (Wuhan, China) between August 2013 and July 2014 after providing written informed consent. None of the patients received chemotherapy or radiotherapy before surgery. The use of tissues for this study was approved by the ethics committee of Tongji Hospital. A total of 39 fresh CRC tissues were frozen in liquid nitrogen until further use. The clinical information included age, gender, stage, tumor size, depth of invasion, lymph node metastasis, and distant metastasis (*Hu et al., 2016*).

## Cell culture

The SW48 and LoVo cell lines, human CRC cell lines, were purchased from the China Center for Type Culture Collection (Wuhan, China). We cultured the cell line at 37 °C in DMEM (Gibco, Thermo Fisher Scientific, Waltham, MA, USA), which contained 10% fetal calf serum (Gibco) under 5% $CO_2$ atmosphere. Cells were collected and analyzed after 72 h of culture.

## Cell transfection

The SIX4-siRNA and control siRNA were constructed by Ruibo Company (Guangzhou, China), and myr-AKT plasmid was a gift from Kira Gritsman (*Kharas et al., 2009*). The siRNA and plasmid were transfected into colorectal cells using the Lipofectamine 2000 reagent (Invitrogen, Carlsbad, CA, USA). Detection of specific mRNAs or proteins using qPCR or western blot was completed 72 h after transfection.

## Quantitative real-time PCR

The Trizol reagent (Invitrogen, Carlsbad, CA, USA) was applied to total RNA extract from cultured cells. PrimeScript™ RT Master Mix (TaKaRa Bio, Otsu, Japan) was used to reverse transcribe the total RNA to cDNA. SYBR Green Real-time PCR Master Mix (TaKaRa Bio, Otsu, Japan) was used to amplify cDNA in a 20 μL reaction system. The primers were designed and synthesized by TsingKe (Wuhan, China). The primer information is as follows: SIX4 upstream 5′-GCATTGAACCCACCAAAAATGT-3′ and downstream 5′-GGAAGTAGACCCCAGTATGTCA-3′, GAPDH upstream 5′-GGAGCGAGATCCCTCCAAAAT-3′ and downstream 5′-GGCTGTTGTCATACTTCTCATGG-3′. GAPDH expression was used for normalization. Finally, we used the ($2^{-\Delta\Delta CT}$) method to calculate the results.

## Western blot analysis

Nonidet P-40 (NP40, Beyotime, Shanghai, China) buffer was used to extract the proteins from colorectal cells. We used the bicinchoninic acid assay to measure protein concentration. The proteins were separated using sodium dodecyl sulfate-PAGE electrophoresis and then transferred to a polyvinylidene difluoride membrane (Millipore, Billerica, MA, USA). Tris-buffered saline with Tween-20 and 5% nonfat milk was used to block the membrane at room temperature. After blocking for 1 h, we incubated with the following primary antibodies at 4 °C overnight: anti-SIX4, anti-T-AKT, anti-P-AKT, and anti-GAPDH (Santa Cruz, CA, USA). GAPDH was used to normalize total protein levels.

## Tumor cell invasion and migration assay

We performed invasion and migration assays using Boyden chambers with or without Matrigel (100 μL, Corning, Shanghai, China) in 24-well dishes. Initially, $1 \times 10^5$ cells in the presence or absence of SIX4 were cultured in the upper chamber. Meanwhile, we placed DMEM containing 20% Fetal Bovine Serum (FBS) in the lower chamber. After 24 h of incubation, cells in the upper chamber were fixed in 4% formaldehyde and stained with 0.05% crystal violet. We counted the number of cells on the lower side of the filters at 200× magnification.

## Scratch-wound assay

We seeded the CRC cells in six-well plates at approximately 100% confluence. A 200 μL pipette tip was used to draw a linear scratch-wound. Next, phosphate buffer solution (PBS) was used to wash the cells gently, and serum-free medium was used to block cell proliferation. After 24 h of culture, the closure area was measured using a microscope by comparing the images from the start time point (0 h) to the last time point (24 h).

## Statistical analysis

R software, SPSS 17.0 statistic software (SPSS), and GraphPad Prism 5.0 software were used to conduct statistical analyses. The R packaged used in the present study were: ggplot2 (*Wickham, 2016*), survival (*Therneau & Grambsch, 2000*), and limma

(*Ritchie et al., 2015*). The data were presented as mean ± SD. Student's *t*-tests were applied to describe the differences between different groups. Kaplan–Meier analysis was used to calculate the survival differences between divided groups. The Pearson correlation test was used to compare the correlation between the SIX4 and AKT pathway genes. We considered $P < 0.05$ as significant in all cases.

## RESULTS

### SIX4 was upregulated in CRC tissues

In order to clarify the expression of SIX4 in CRC tissues, we calculated the RNA-sequencing data from TCGA database. We extracted 380 CRC samples and 50 normal colorectal samples from TCGA data portal. The expression of SIX4 was substantially higher in CRC tissues than in normal tissues (Fig. 1A). In addition, we investigated SIX4 expression in four other cohorts: GSE5206, GSE20916, GSE6988, and GSE20842. These results were consistent with TCGA CRC cohort (Figs. 1B–1E). Moreover, qPCR detected SIX4 mRNA expression in 12 pairs of frozen CRC samples. The results demonstrated that SIX4 mRNA levels were upregulated in CRC tissues compared to those in matched normal colorectal tissues, which confirmed the above results (Fig. 1F). We investigated SIX4 protein levels in CRC tissues and their adjacent normal controls by western blot and calculated the relative protein expression via gray scanning. The results showed an increase in SIX4 protein expression in CRC tissues (Fig. 1G).

### Upregulation of SIX4 was associated with poor CRC survival

We divided the colorectal patients into two groups according to their SIX4 expression levels. Interestingly, the overall survival results showed that patients with high SIX4 expression had significantly worse outcomes in TCGA CRC database (Fig. 2A). In addition, Cox regression was used for multivariate survival analysis, including TNM stage, location, gender, and age. The results showed a strong correlation between SIX4 upregulation and poor survival (Table 1) The relapse-free survival result also revealed a poor prognosis in the patients with high SIX4 expression (Fig. 2E). Moreover, we further analyzed the association between SIX4 expression level and CRC prognosis in the following CRC cohorts: GSE39582, GSE17536, and GSE14333. Notably, patients in the SIX4-high group had poor survival in all cohorts (Figs. 2B–2D). We analyzed the relapse-free survival data in the GSE39582 and GSE17536 cohorts, which confirmed the results from TCGA CRC cohort (Figs. 2F and 2G).

### SIX4 mRNA level was related to TNM stage and lymph node metastasis

In order to clarify the effect of SIX4, we further analyzed the relationship between SIX4 mRNA levels and different clinicopathological features of CRC patients. There was no significant association between SIX4 expression and gender, age, or metastasis distance (Table 2). However, the SIX4 mRNA level significantly correlated with lymph node metastasis (Fig. 3A) and TNM stages (Fig. 3B). Other CRC patient cohorts, including GSE5206, GSE14333, and GSE39582, confirmed this result. The results from GSE5206 and

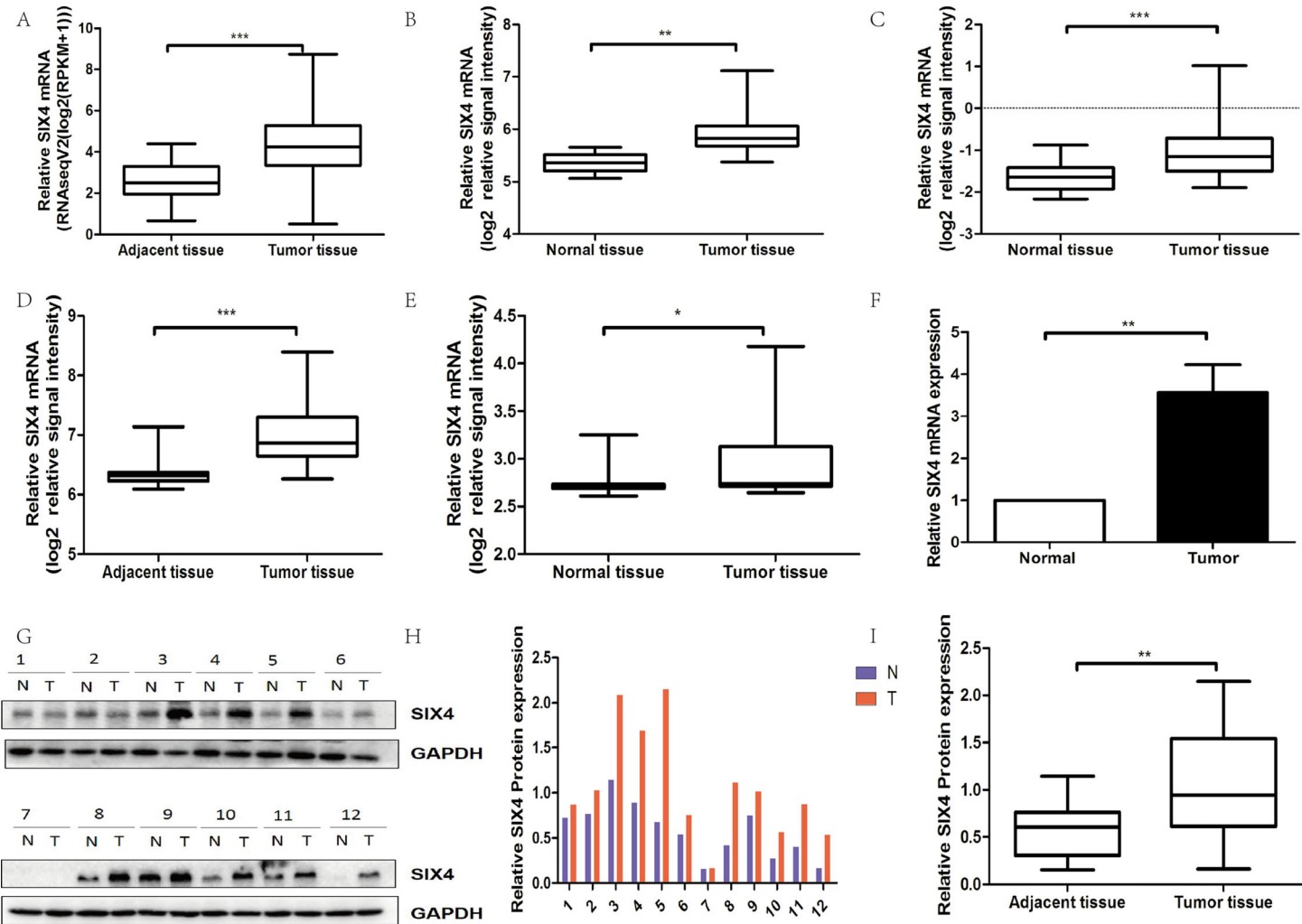

**Figure 1 SIX4 was upregulated in CRC tissues.** (A) TCGA CRC cohort was used to calculate the expression of SIX4 in colorectal cancer (CRC) and normal colorectal tissues. (B–E) GEO cohorts (GSE5206 (B), GSE6988 (C), GSE20842 (D), and GSE20916 (E)) were used to calculate the expression of SIX4 in CRC and normal colorectal tissues. (F) qPCR assay was used to analysis the mRNA expression of SIX4 in CRC and their matched control tissue. (G) Western blot assay was used to analysis the protein expression of SIX4 in colorectal cancers and their matched control tissues. (H) Quantification of SIX4 protein in colorectal tissues. Band intensities were measured using Image J and normalized to GAPDH. (I) Quantification of SIX4 is shown by box-whisker plot. ($^*P < 0.05$, $^{**}P < 0.01$, and $^{***}P < 0.001$).

GSE39582 showed higher SIX4 expression in lymph node metastasis samples compared with that in negative metastasis samples (Figs. 3C and 3D). In addition, the GSE14333 cohort showed SIX4 overexpression in Dukes C and D stages, which have invaded the lymph nodes, compared to Dukes A and B stages, which have no lymph node metastasis (Fig. 3E). Moreover, we examined SIX4 expression in 39 CRC tissues with complete clinical information (*Hu et al., 2016*). SIX4 expression was higher in lymph node metastasis tissues than in tissues without lymph node metastases (Fig. 3F). Finally, the correlations between SIX4 expression and clinicopathological features showed a similar conclusion (Table 3). These results demonstrate that SIX4 expression is related to TNM stage and lymph node metastasis.

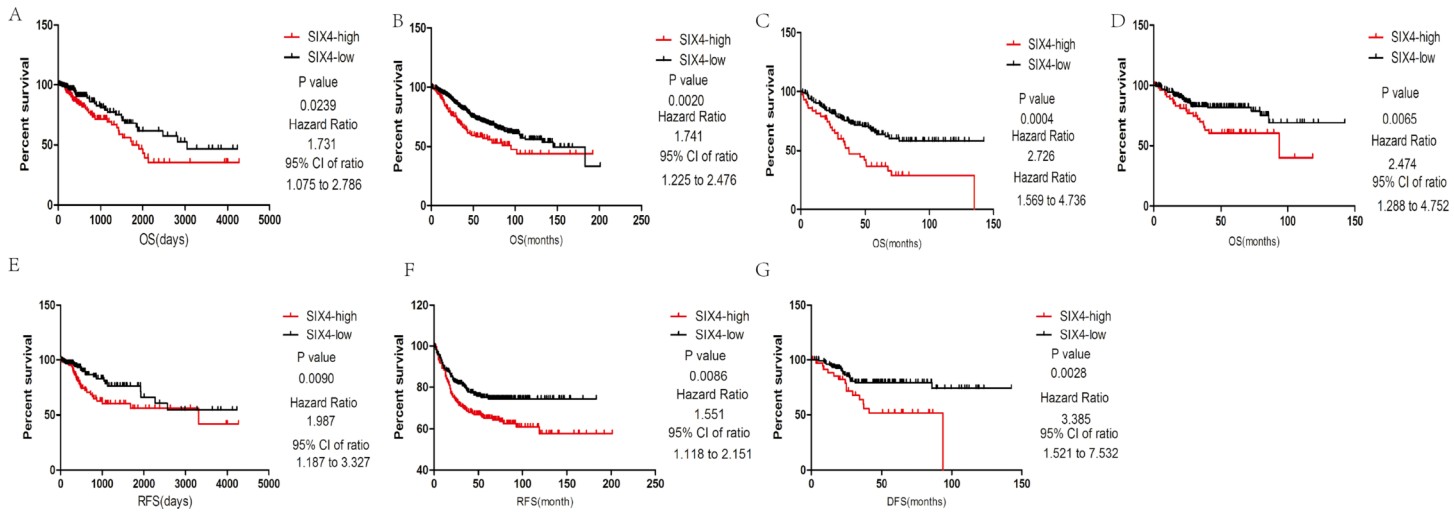

**Figure 2 Upregulation of SIX4 was associated with poor CRC survival.** (A–D) Overall survival analysis showed that high expression of SIX4 in tumors correlated with a poor prognosis in TCGA(A) and GEO cohorts (GSE39582 (B), GSE17536 (C), and GSE14333 (D)). (B) Relapse-free survival (RFS) and disease-free survival (DFS) analysis showed that tumors with higher expression of SIX4 have a poor prognosis in TCGA(E) and GEO cohorts (GSE39582 (F) and GSE17536 (G)).

**Table 1 Multivariate cox analysis of overall survival in TCGA CRC cohort.**

|  | B | P value | Exp (B) | 95.0% Exp (B) | |
|---|---|---|---|---|---|
| SIX4 | 0.174231 | 0.037416 | 1.19033 | 1.010201 | 1.402579 |
| Location | −0.0409 | 0.41426 | 0.959926 | 0.870155 | 1.058959 |
| TNM stage | 0.609429 | 8.63E-05 | 1.83938 | 1.356896 | 2.493426 |
| Gender | 0.297522 | 0.25709 | 1.346519 | 0.804911 | 2.252564 |
| Age | 0.753558 | 0.009382 | 2.124546 | 1.203245 | 3.751268 |

## SIX4 knockdown inhibited invasion and migration in CRC cells

To further study the function of SIX4 in CRCs, SW48, and LoVo cells were transfected with siRNAs specific for SIX4 and control siRNA. The number of invaded cells significantly decreased in SW48 and LoVo cells expressing SIX4 siRNA than in the control cells (Figs. 4A and 4B). In addition, wound healing assays showed SIX4 knockdown decreased the migration ability of SW48 and LoVo cells. After 24 h, cells with SIX4 siRNA migrated longer distances than cells with control siRNA (Fig. 4C). Thus, silencing SIX4 inhibited tumor cell invasion and migration capacity.

## PI3K-AKT pathway was regulated by SIX4 in CRC

Our results demonstrated that SIX4 expression level was related to lymph node metastasis and SIX4 knockdown inhibited this invasion and migration capability. However, the molecular mechanisms were still undefined. To further reveal the molecular mechanisms of SIX4 in CRC, GO, and KEGG enrichment analyses were carried out. First, we used Pearson correlation to calculate the co-expression of SIX4 with other genes. There were 2,093 genes that correlated with SIX4 (Pearson $r > 0.3$ or $< −0.3$) (Fig. 5A). Next, DAVID Bioinformatics tools were used for GO and KEGG enrichment analysis

**Table 2 The relationship of expression of SIX4 with clinical feature of TCGA.**

| Clinical feature | | SIX4 (mean ± SD) | t | P |
|---|---|---|---|---|
| Age | <60 | 4.385 ± 1.483 | 0.117 | 0.907 |
| | ≥60 | 4.365 ± 1.553 | | |
| Sex | Male | 4.312 ± 1.477 | 0.817 | 0.414 |
| | Female | 4.444 ± 1.588 | | |
| M stage | M0 | 4.366 ± 1.52 | 0.117 | 0.907 |
| | M1 | 4.387 ± 1.554 | | |
| T stage | T1 + T2 | 3.587 ± 1.372 | 4.857 | 1.889E-06 |
| | T3 + T4 | 4.556 ± 1.506 | | |
| TNM stage | I | 3.704 ± 1.398 | 3.664 | 2.857E-04 |
| | II + III + IV | 4.499 ± 1.52 | | |
| N stage | N0 | 4.007 ± 1.557 | 2.33 | 0.02 |
| | N1 + N2 | 4.365 ± 1.554 | | |

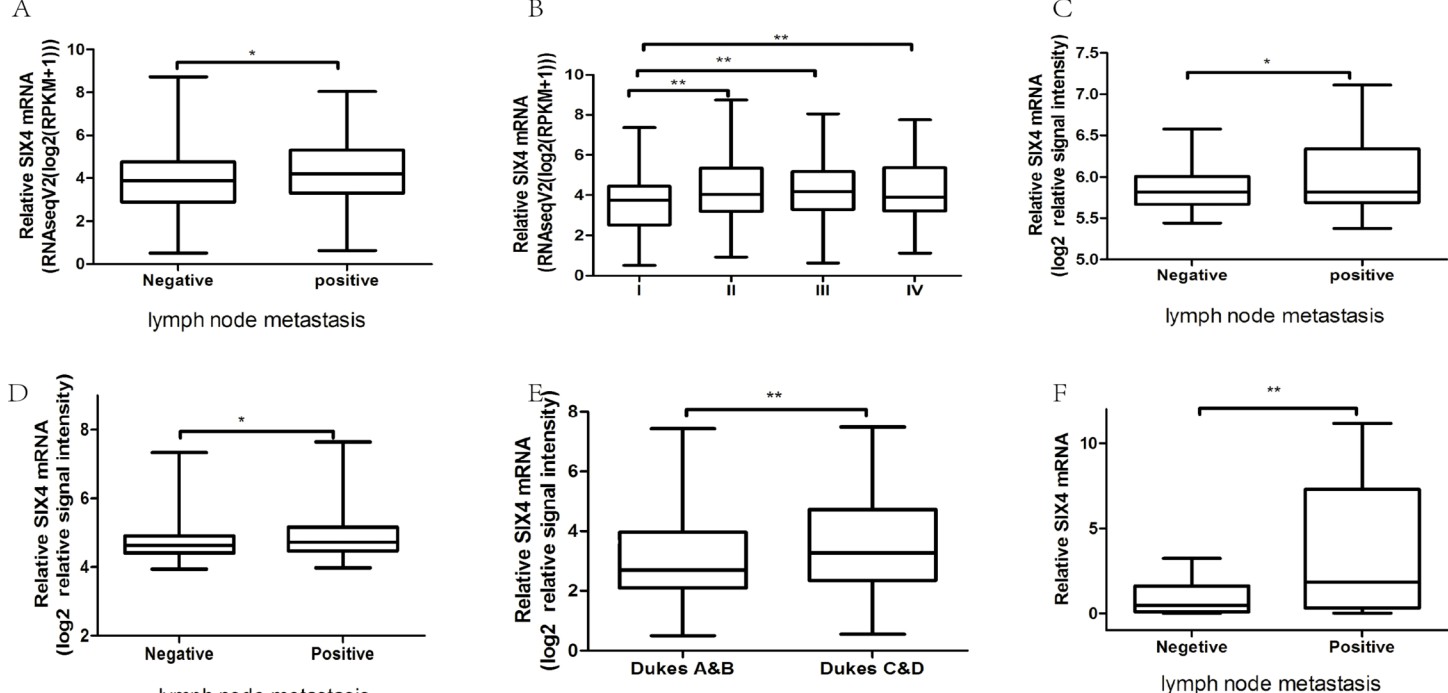

**Figure 3 SIX4 mRNA level was related to TNM stage and lymph node metastasis.** (A,B) The expression of SIX4 in different lymph node stages and TNM stages of TCGA cohort. (C,D) The expression of SIX4 in different lymph node stages in GEO cohorts (GSE5206 (C) and GSE39582 (D)). (E) The expression of SIX4 in different Dukes stages in GEO cohort (GSE14333). (F) The expression of SIX4 in different lymph node stages in colorectal cancer tissues (*P < 0.05, **P < 0.01).

(*Huang, Sherman & Lempicki, 2008*). The results from the GO analysis for co-expressed genes showed that the gene functions significantly focused on several GO terms including cell adhesion, biological adhesion, extracellular structure organization, and blood vessel development. Cellular component terms associated with the SIX4-related gene included extracellular matrix (ECM), proteinaceous ECM, and extracellular

**Table 3 The relationship of expression of SIX4 with clinical feature of colorectal cancer.**

| Clinical feature | | SIX4 (mean ± SD) | t | P |
|---|---|---|---|---|
| Age | ≥60 | 1.607 ± 1.824 | 1.062 | 0.295 |
| | <60 | 2.651 ± 3.771 | | |
| Sex | Male | 1.282 ± 2.501 | 1.36 | 0.182 |
| | Female | 2.98 ± 3.569 | | |
| M stage | M0 | 1.87 ± 2.704 | 0.599 | 0.553 |
| | M1 | 2.913 ± 3.918 | | |
| T stage | T1 + T2 | 0.721 ± 0.934 | 0.977 | 0.335 |
| | T3 + T4 | 2.305 ± 3.143 | | |
| TNM stage | I + II | 0.790 ± 1.265 | 2.915 | 0.006 |
| | III + IV | 3.558 ± 3.849 | | |
| N stage | N0 | 2.038 ± 2.886 | 3.177 | 0.003 |
| | N1 + N2 + N3 | 3.751 ± 3.875 | | |

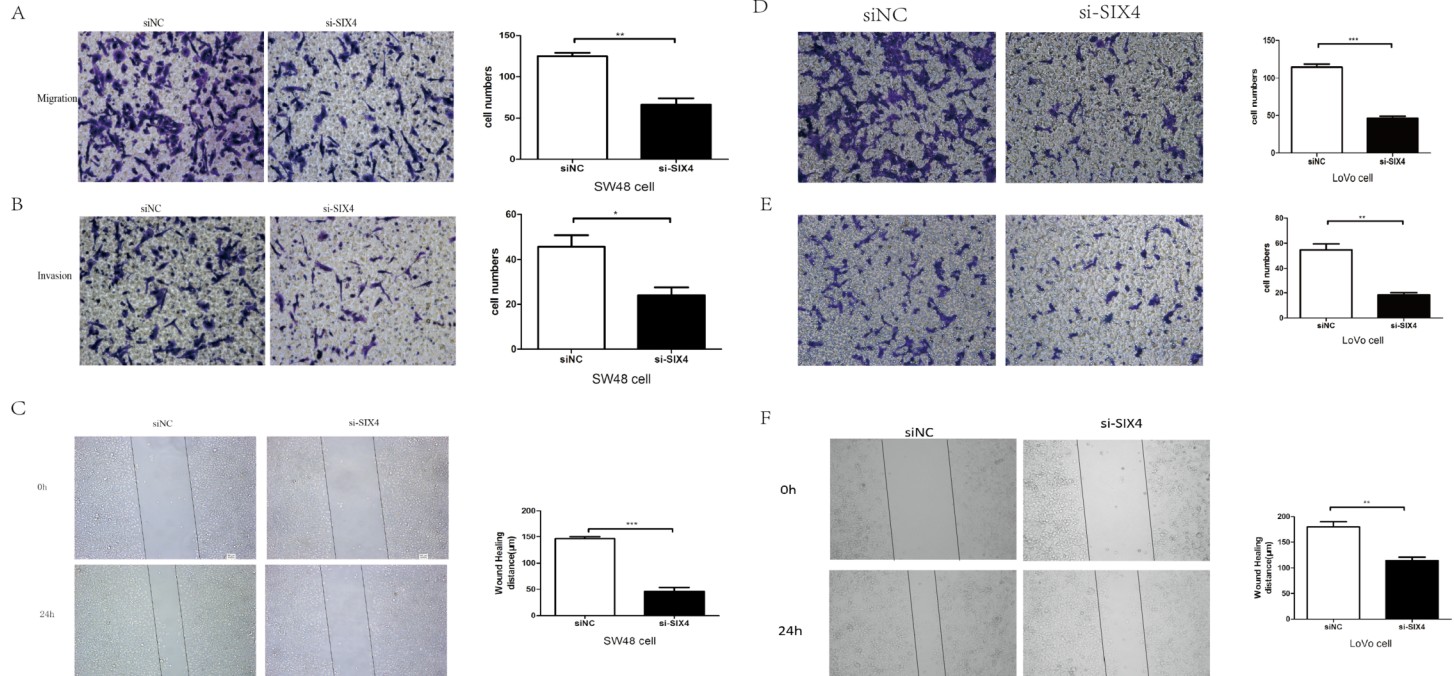

**Figure 4 SIX4 knockdown inhibited invasion and migration in CRC cells.** (A,B) Cell migration and invasion assays were used to study the migration and invasion ability in SW48 cells with siNC or si-SIX4. (C) Scratch-wound assay was used to study the migration ability of SW48 cells with siNC or si-SIX4. (D,E) Cell migration and invasion assays were used to study the migration and invasion ability in LoVo cells with siNC or si-SIX4. (F) Scratch-wound assay was used to study the migration ability of LoVo cells with siNC or si-SIX4 ($*P < 0.05$, $**P < 0.01$, and $***P < 0.001$).

structure. Molecular function terms associated with SIX4 associated genes included calcium ion binding, polysaccharide binding, pattern binding, and glycosaminoglycan binding (Fig. 5B). KEGG pathway analysis provided significant pathway information for SIX4 co-expressed genes, including ECM-receptor interaction, focal adhesions, PI3K-AKT signaling pathway, and osteoclast differentiation (Fig. 5C). In addition,

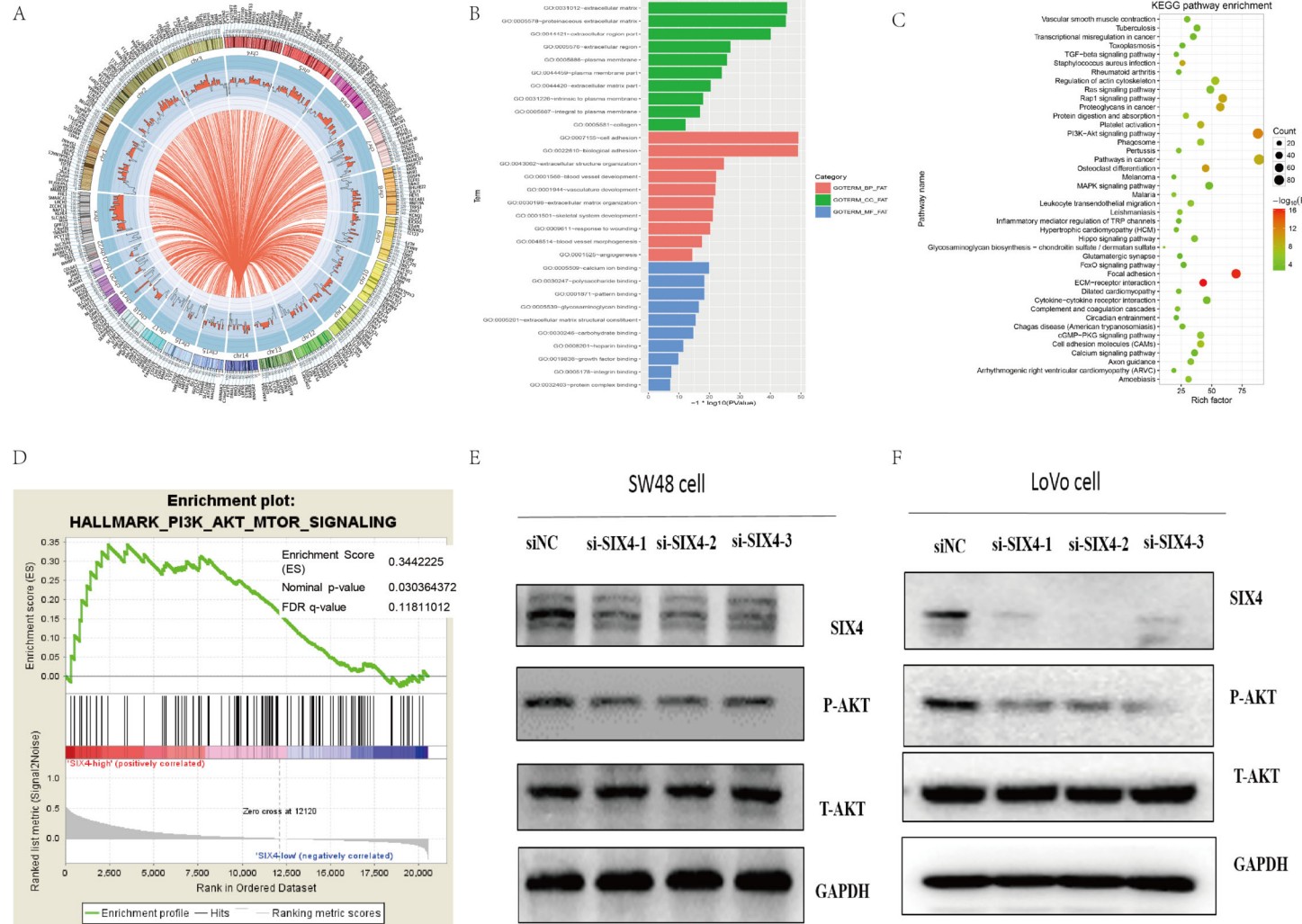

**Figure 5 PI3K-AKT pathway was regulated by SIX4 in CRC.** (A) Circos plot shows 2,093 genes co-expressed with SIX4. The histogram shows the Pearson correlation coefficients between SIX4 and the co-expressed genes, with red color representing SIX4 positive co-expression genes and blue indicating negative co-expression genes. (B) Gene ontology (GO) and (C) Kyoto encyclopedia of genes and genomes (KEGG) enrichment analysis identified significant terms of SIX4 co-expression genes in TCGA colorectal cancer dataset. (D) Gene set enrichment analysis (GSEA) identified significant associations between SIX4 co-expression genes and the PI3K-AKT signaling pathway in TCGA CRC cohort. (E,F) SW48 and LoVo cells were transfected by si-SIX4 or control siRNA, followed by western blot analysis using specific antibodies for SIX4, phosphorylated AKT (pAKT), or total AKT.

we performed GSEA analysis using RNA-sequencing data from the CRC cohort of TCGA (380 patients). Among the "hallmark signature" gene sets, the PI3K-AKT pathway had a relationship with SIX4 expression levels in TCGA CRC dataset. The nominal *P* value of PI3K-AKT pathway was <0.05, however, the FDR *Q* value and FWER *P* value were >0.05 (Fig. 5D). Finally, western blot assays were performed for SW48 and LoVo cells to further validate the above bioinformatics results. The gray scan results showed that p-AKT was significantly decreased when SIX4 was silenced in SW48 and LoVo cells (Fig. 5E).

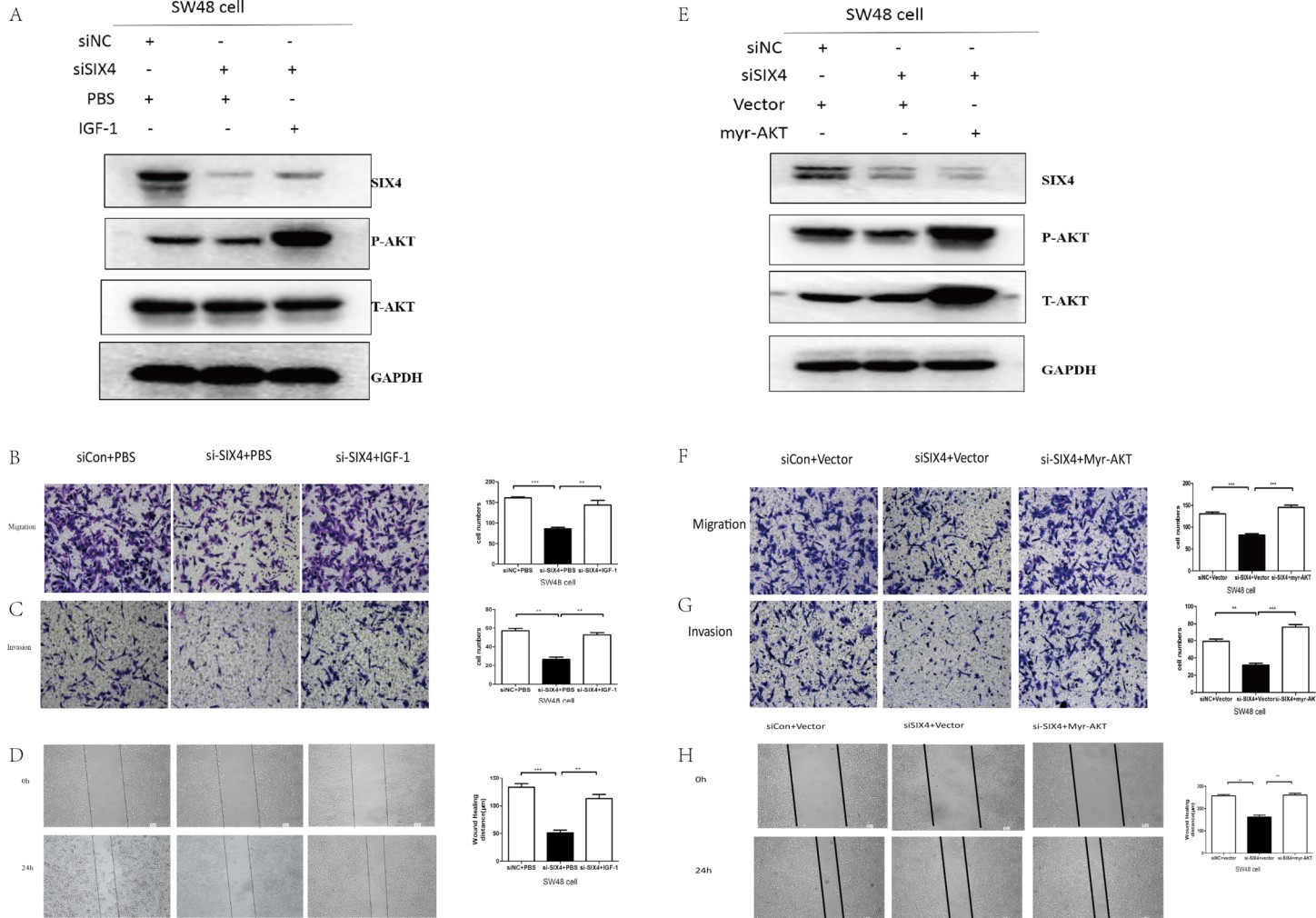

**Figure 6 SIX4 promoted metastasis via the PI3K-AKT pathway.** (A) SW48 cells were transfected by siNC, si-SIX4, or si-SIX4 with IGF (100 nM), followed by western blot analysis using specific antibodies for SIX4, phosphorylated AKT (pAKT), or total AKT. (B,C) Cell migration and invasion assays were used to analysis the ability of SW48. (D) Wound-healing assay was used to analysis the migration ability of SW48 cells. (E) SW48 cells were transfected by siNC, si-SIX4, or si-SIX4 with myr-AKT, followed by western Blot analysis using specific antibodies for SIX4, pAKT, or total AKT. (F,G) Cell migration and invasion assays were used to analysis the ability of SW48. (H) Wound-healing assay was used to analysis the migration ability of SW48 cells ($^{*}P < 0.05$, $^{**}P < 0.01$, and $^{***}P < 0.001$).

## SIX4 promoted metastasis via the PI3K-AKT pathway

To confirm whether the effects of SIX4 on CRC cells is dependent on PI3K-AKT signaling, we simultaneously decreased SIX4 expression levels and activated the AKT signaling pathway. IGF-1 and myr-AKT were used to carry out the rescue assay. Notably, the western blot assay confirmed that IGF-1 and myr-AKT increased p-AKT expression, a key marker of PI3K-AKT (*Kharas et al., 2009*; *Ohta et al., 2015*; *She et al., 2005*). In addition, the transwell assay showed the migrating and invading cells decreased during SIX4 silencing, whereas the effect was reversed when the AKT signaling pathway was activated by IGF-1 (Figs. 6A–6D) or myr-AKT (Figs. 6E–6H).

## DISCUSSION

SIX4 is a member of the homeobox family required for eye development (*Liu et al., 2015*). SIX4 is a transcription factor and may participate in neuronal cell differentiation or maturation (*Santolini et al., 2016*). SIX4 can act as both a transcriptional repressor and activator through binding DNA sequences on its target genes. There is increasing evidence that SIX family members are not only correlated with regulating precursor cell proliferation and differentiation, but also contribute to oncogenesis (*Xu et al., 2016*). SIX4 is expressed in esophageal squamous cell carcinoma (*Wei et al., 2013*). However, the SIX4 expression in CRC tumors is unknown. In this study, we verified SIX4 expression in CRC and examined the association between clinical features and SIX4 expression in CRC. Our results show that both SIX4 mRNA and protein levels were significantly higher in CRC tissues than in control tissues. We validated the conclusion though several large cohorts of CRC patients from TCGA and GEO databases, revealing that SIX4 expression was related to CRC development.

The SIX family contains evolutionarily conserved transcription factors that play important roles in cell proliferation, differentiation, apoptosis, adhesion, and migration (*Christensen, 2007*; *Li et al., 2003*; *Mo et al., 2013*). We found that SIX4 levels might have a predictive effect for CRC patient prognosis because patients with high levels of SIX4 had a poor prognosis in TCGA CRC cohort. Results from several cohorts, including GSE39582, GSE17536, and GSE14333, verified this finding. Moreover, our study demonstrated that SIX4 expression correlated with lymph node metastasis and TNM stage in TCGA database, which was validated in the GSE5206, GSE14333, and GSE39582 cohorts. Therefore, SIX4 may be related to CRC lymph metastasis and predict CRC patient prognosis. In addition, we investigated SIX4 function by silencing SIX4 in CRC cells. The results show that knockdown of SIX4 can inhibit cell migration and invasion in SW48 and LoVo cells.

However, the mechanisms that SIX4 may regulate in cancer progression remained unclear. In order to clarify the SIX4-associated pathways, bioinformatics analysis was applied using TCGA RNA-sequencing data. The bioinformatics analysis included GO, KEGG, and GSEA. First, we selected the SIX4 co-expression genes via Pearson correlation analysis yielding 2,283 genes considered SIX4 co-expression genes ($r > 0.3$ or $r < -0.3$). The results of the GO analysis with co-expression genes showed that the genes significantly focused on several GO terms including cell adhesion, biological adhesion, extracellular structure organization, and blood vessel development. The cellular component terms associated with SIX4-related genes were involved in ECM, proteinaceous ECM, and extracellular structure. The molecular functions for co-expressed genes included calcium ion binding, polysaccharide binding, pattern binding, and glycosaminoglycan binding. KEGG pathway analysis provided significant pathway information for SIX4 co-expressed genes, including ECM-receptor interaction, focal adhesions, PI3K-AKT signaling pathway, and osteoclast differentiation. GSEA analysis also demonstrated that the PI3K-AKT pathway might have an effect on SIX4's function in CRC. PI3K-AKT signaling positively correlated with CRC progression. Several studies

demonstrate that the PI3K-AKT signaling pathway is involved in the metastasis process (*Dey et al., 2010*; *Fang et al., 2012*). Here, the results of the western blot assay showed that SIX4 knockdown inhibited the AKT pathway. IGF-1 and myr-AKT were applied to activate the AKT pathway when SIX4 was silenced in SW48 cells (*Stitt et al., 2004*). Several studies show that IGF-1 and myr-AKT functioned as AKT activators (*Calvisi et al., 2011*; *Kharas et al., 2009*; *Sujobert et al., 2015*). The migration and invasion assays showed that the effect caused by silencing SIX4 was reversed when the AKT signaling pathway was activated via IGF-1 and myr-AKT. Together, these data prove that SIX4 affected the PI3K-AKT pathway and then contributed to CRC metastasis.

## CONCLUSION

In conclusion, the present study demonstrated that SIX4 overexpression in CRC tissues correlated with lymph node metastasis and predicted poor prognosis in CRC patients. In addition, bioinformatics analyses, including GO, KEGG, and GSEA analysis, showed that SIX4 induces the PI3K-AKT pathway, which was confirmed by western blot assay. Our research contributes to finding new prognosis biomarkers for CRC and provides a novel molecular mechanism for CRC lymph metastasis.

### Funding
This study was supported by the National Natural Science Foundation (Nos. 81372323 and 81372662). The funders had no role in study design, data collection and analysis, decision to publish, and or preparation of the manuscript.

### Grant Disclosures
The following grant information was disclosed by the authors:
The National Natural Science Foundation: 81372323 and 81372662.

### Competing Interests
The authors declare that they have no competing interests.

### Author Contributions

- Guodong Li conceived and designed the experiments, performed the experiments, analyzed the data, contributed reagents/materials/analysis tools, wrote the paper, prepared figures and/or tables, and reviewed drafts of the paper.
- Fuqing Hu performed the experiments, prepared figures and/or tables, and reviewed drafts of the paper.
- Xuelai Luo analyzed the data, contributed reagents/materials/analysis tools, and reviewed drafts of the paper.
- Junbo Hu contributed reagents/materials/analysis tools, wrote the paper, and reviewed drafts of the paper.
- Yongdong Feng conceived and designed the experiments, wrote the paper, prepared figures and/or tables, and reviewed drafts of the paper.

## Human Ethics

The following information was supplied relating to ethical approvals (i.e., approving body and any reference numbers):

Our study was approved by the Ethical Committee of Tongji Hospital, Tongji Medical College, Huazhong University of Science and Technology (Ethical Application Ref: TJ-C20150805).

## Data Availability

The raw data has been supplied as Supplemental Dataset Files.

## Supplemental Information

Supplemental information for this article can be found online at http://dx.doi.org/10.7717/peerj.3394#supplemental-information.

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
