# Peer review of "SIX4 promotes metastasis via activation of the PI3K-AKT pathway in colorectal cancer"

_PeerJ, doi:10.7717/peerj.3394_

## Round 0.1 · original submission · Major Revisions

· Academic Editor

Major Revisions

Please address the reviewer comments in a point-by-point fashion. Thank you for your interest in PeerJ!

·

Basic reporting

Overall, this paper clearly reported the expression and function of SIX4 gene, which was of great clinical significance. Several concerns still need to be addressed.
1. The language should be revised by a native speaker.
2. Some of the expression is not clear. For example, in western part, it is hard to know which sample did the author use.

Experimental design

1. In validation stage, the author only use six pairs of samples, it is too small to draw a conclusion.
2. Also, it will be better if the author use their own patient corhort to compare the expression with clinical parameters.
3. In the study of function analysis, the author use only one cell line.
4. In the study of PI3k-Akt pathway, why the author just use western blot to validate the result?

Validity of the findings

no comment

Additional comments

Overall the paper is comprehensive and well-designed, using public available data.

Reviewer 2 ·

Basic reporting

no commen

Experimental design

no commen

Validity of the findings

no commen

Additional comments

This manuscript uses both publically available repository and their cell-culture data to examine the importance of SIX4 in promoting lymph node metastasis in colorectal cancers. Several GEO datasets and TCGA cases were used. Their findings are interesting, and appear supported in part by their data. However, there are some major concerns.

Major points
1. Multivariate analyses are recommended to rule out potential confounding if possible.
2. The baseline characteristics of the included cases should be shown in a table and ideally compared between the SIX4 + and - groups. This would be very challenging for the publically valiable data/repositories.
3. Data is not so robust to prove your result, such as “SIX4 promotes lymph metastasis via the PI3K-AKT pathway” (Line 199).
4. Grammatical errors and writing style issues.

Minor points
5. Lines 10-11. I suggest you couldemphasize the importance of SIX4 at first, and then is the innovationof you study.
6. The English language could be improved to ensure that your international audience can clearly understand your text. I suggest that you have a native English speaking colleague review your manuscript. Some examples where the language could be improved include lines 14,141, 174-175.
7. In line 134, “SIX4 was upregulation in CRC tissues”, I suggest “upregulation” could be changedto “upregulated”.
8. I thank you for sharing your own in vitro data. When you describe your results, the tenses need to be consistent.It is recommended to use the general past tense (Lines 13-18). I also suggest you to describe the specificresults (Lines 13-14).
9. Line 171 “The function of SIX4 in CRC”. Maybe this title is ambiguous, not specific cell function. So as to the title “The relationship of SIX4 expression and CRC patients prognosis“(Line 147).and “The molecular mechanisms of SIX4 in CRC” (Line 177). I suggest you describe the specific molecular mechanisms invovled in you study, such as “Silence of SIX4 inhibited invasion and migration in CRC”.
10. “According to SIX4 expression levels, we divided the colorectal patients into two groups” (Line 147), I suggest youto give the specific grouping criterion.
11. It is suggested that the gray scale value could be marked Figure 1G, because not all SIX4s in cancer tissues are highly expressed, compared to the adjancent cancer tissues. Meantime, the quality of layout in Fig 4 could be improved.

Reviewer 3 ·

Basic reporting

In this study, Li et al investigated the role of SIX4 in progression and prognosis of colorectal cancer using bioinformatics analysis of existing online databases and several in vitro experiments. They found that SIX4 was generally increased in TCGA and GEO cohorts. They also observed that the higher level of SIX4 was associated with poor prognosis and metastasis toward lymph nodes. Mechanistically, they attempted to demonstrate the effect of SIX4 on colorectal cancer cell migration and invasion is dependent on PI3K-AKT signaling.

Overall, the language of this draft is poor, many spelling and grammatical errors. Many sentences are not described clearly, some confusing and overinterpreted. Given these craftsmanship issues, it is difficult to properly evaluate the experiments and results .

Experimental design

Experiments in this study are superficially described. More detailed information need to be provided.

Validity of the findings

The data from in vitro experiments did not help explain why the patients with higher SIX4 exhibit higher frequency of lymph node metastasis. Did these cancer cells express more homing receptors?

In Fig.5C, Akt phosphorylation was not significantly reduced in SIX4 knockdown lanes.

IGF treatment can activate several pathways besides PI3K-AKT. Without proper controls and rigorous experiment design, the authors cannot draw the conclusion that the effect of SIX4 is PI3K-AKT signaling dependent.

Additional comments

This draft need to be thoroughly revised by either English native speaker or professional English editing company. Some conclusions need to be tuned down since supporting evidences are not that strong.

Annotated reviews are not available for download in order to protect the identity of reviewers who chose to remain anonymous.

Reviewer 4 ·

Basic reporting

The language is unambiguous and professional, except some passages (for example the Abstract). Minor grammar mistakes should be revised.

The field background is understandably low, due to the poor scientific background of SIX4 in the literature.

The article is well structured.

The abstract should be revised: please explain SIX4, the conclusion is not meaningful.

Experimental design

no comment

Validity of the findings

Major revision:
Only one cell line was used in the experiments. The authors should analyze at least one other colorectal cell line for valid statements.

Additional comments

Please check and complete the abbreviations.

The reference for “More importantly, it is closely linked to a poor clinical prognosis in cancer patients. Furthermore, SIX2…” (line 40/41 in Word document) is missing.

The aim of this study is not clearly worked out (line 43-46).

Tumor cell invasion and migration assay: which pore diameter was used?

Please add a legend in Fig 1 A-E for the x-axis.

Figure 1G: The authors wrote: “We investigated SIX4 protein levels in colorectal cancer tissues and their adjacent normal controls by western blot and found an increase in SIX4 protein expression in colorectal cancer tissues (Fig. 1G).” Only 3 of the 6 examined cancer tissue probes showed a clear increased SIX4 level. The results are not significant enough. More probes should be investigated.
Are there any differences between the human probes with increased SIX4 expression and the probes with no increased expression, which could explain these different results?

Figure 2: Did you analyze all 8 cohorts or only the six cohorts? If yes, why did you analyze or show only these 6 cohorts?

Figure 4: How did you measure and analyze the closure area in detail?

Figure 5E: the authors wrote: “And then we found that p-AKT was significantly decreased when SIX4 was silenced (Fig. 5E).” Was significance investigated in western blot analysis? How was it investigated? How often was this experiment independently repeated for a valid statement?
Figure 4 and 6: How many independent repetitions of the experiments were performed?

Supplement:
Western blot: Please explain the gel loading, which probes were analyzed?

---

## Round 0.2 · Minor Revisions

· Academic Editor

Minor Revisions

Overall, the manuscript has been significantly improved, while some minor concerns remained outstanding. Thank you for your thorough and careful responses!

1. Is Dr. Guodong Li a co-corresponding author? If not, please correct this in the PeerJ online system which shows that he or she is.
2. There are at least 19 known and widely used colorectal cancer datasets in GEO. How and why did the authors pick the 6 used in the study? This selection may lead to a selection bias. Please specify and justify this selection process in the method section.
3. In general, there is no statistically difference, when two survival curves cross or overlap in Kaplan-Meier plot. Please double check that the p values in Fig 2B and 2E are corrected. Ideally, both logrank and Cox proportional regression analyses should be used and agreed on their respective results.
4. Please specify what packages of R was used for the statistical analysis since R-project itself alone cannot be sufficient to conduct all findings presented here.
5. Fig. 5D, please also report normalized enrichment score (NES and familywise error rate (FWER). If FDR's q and FWER's p values are both >0.05, the difference between SIX-high and -low groups would not be statistically significant.
6. Fig. 3: N staging would not be positive or negative, but nodal (lymph node) metastasis could. Please change the x-axis labels accordingly. Fig 3D lacked x-axis label.
7. Please explain the meaning of asterisks (** and *) in Fig. 1, 3, 4 and 6 in figure legends, which may indicate statistical significance.
8. Table 1, please follow the AMA style for reporting p values. If not, please at least replace the p value of 8.65E-05 to <0.001. All “p”s in the p-values should be in italics.
9. In abstract, please replace cancer formation to cancer development if agreed. Also, please replace “compared to” to “compared using.” I feel “can act” is less desirable than “can be used.”

---

## Round 0.3 · accepted · Accept

· Academic Editor

Accept

Please consider the annotated manuscript (see attached) for your final edition, in which I attempted to correct some trivial grammtical errors. Thank you!